# HandsOnVLM: Vision-Language Models for Hand-Object Interaction Prediction

**Chen Bao**                                                    *chenbao@cmu.edu*
*Carnegie Mellon University*

**Jiarui Xu**                                                    *jix026@ucsd.edu*
*UC San Diego*

**Xiaolong Wang**[†]                                             *xiw012@ucsd.edu*
*UC San Diego*

**Abhinav Gupta**[†]                                             *abhinavg@cs.cmu.edu*
*Carnegie Mellon University*

**Homanga Bharadhwaj**[†]                                        *homangablackhole36@gmail.com*
*Carnegie Mellon University*

**Reviewed on OpenReview:** *https://openreview.net/forum?id=ehhMFjKnWm*
[†] *denotes equal advising.*

## Abstract

How can we predict future interaction trajectories of human hands in a scene given high-level colloquial task specifications in the form of natural language? In this paper, we extend the classic hand trajectory prediction task to several tasks involving explicit and implicit language queries. Our proposed tasks require an extensive understanding of human daily activities and reasoning abilities about what is happening next given cues from the current scene. We also develop new benchmarks to evaluate the proposed two tasks, Vanilla Hand Prediction (VHP) and Reasoning-Based Hand Prediction (RBHP). We enable solving these tasks by integrating high-level world knowledge and reasoning capabilities of Vision-Language Models (VLMs) with the auto-regressive nature of low-level ego-centric hand trajectories. Our model, *HandsOnVLM* is a novel VLM that can generate textual responses and produce future hand trajectories through natural-language conversations. Our experiments show that *HandsOnVLM* outperforms existing task-specific methods and other VLM baselines on proposed tasks, and demonstrates its ability to effectively utilize world knowledge for reasoning about low-level human hand trajectories based on the provided context. More details can be found at https://www.chenbao.tech/handsonvlm/.

## 1 Introduction

Humans interact with the everyday world and express themselves with informal and oftentimes vague language descriptions. Consider the example in Fig. 1 - when we try to open the jar, we might think, "Ah, I need something to help open this slippery jar more easily." We are uncertain about *what* we exactly want as well as about *how* to come up with a solution. Building a computational system addressing this need would require a good understanding of what tools we have lying around (visual scene understanding), general apriori experience of opening jars (reasoning ability and world knowledge priors), and the ability to actually execute the necessary actions for opening the jar (low-level trajectory). In this paper, we develop language-conditioned prediction tasks for tackling this problem, propose benchmarks for evaluating progress

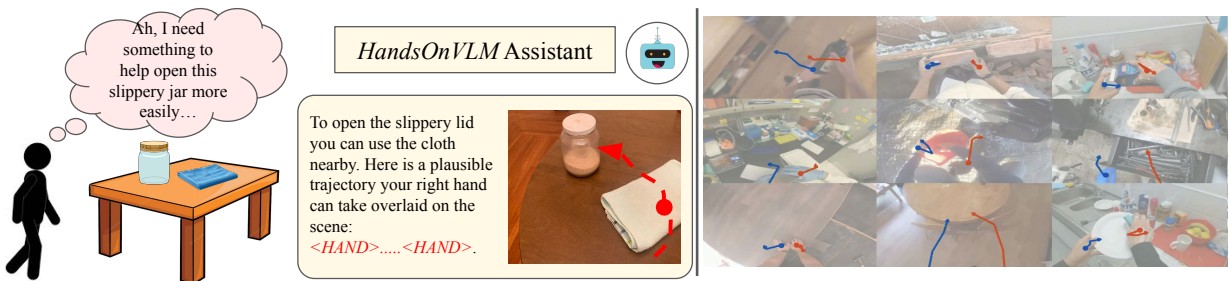

Figure 1: *HandsOnVLM* forecasts low-level actions in the form of hand trajectories in the user's egocentric view of a scene when queried with a question via natural language. It is also capable of handling indirect queries that require reasoning about *what* object to interact with and *how* to perform the interaction. [Right] We show a glimpse of left and right hand trajectory predictions from *HandsOnVLM* across diverse real-world scenarios.

on these tasks, and build a vision-language model (VLM) for predicting low-level hand trajectories in a user's egocentric view of a scene given colloquial language queries.

Towards a similar goal, some prior works have focused on identifying human intentions based on egocentric human videos of daily activities (high-level intentions of the form "cutting pepper", "washing plates") (Krishna et al., 2017; Grauman et al., 2022; Kahatapitiya et al., 2024), while others have focused on predicting low-level actions such as hand trajectories given human action clips (Liu et al., 2022; Zhang et al., 2024b) without conditioning the predictions on detailed language descriptions of the task to be performed. Both these scenarios are a bit restrictive since for most everyday tasks (e.g. in Fig. 1) we need a combination of high-level reasoning of what to do in a scene and low-level understanding of how to interact with the relevant objects in the scene.

By drawing on the recent successes of VLMs for high-level reasoning (Liu et al., 2024; Lai et al., 2024; Cheng et al., 2024) and advancements in hand reconstructions from generic web videos (Shan et al., 2020b; Rong et al., 2020; Pavlakos et al., 2024), we develop a system for future hand trajectory prediction given conversation-style language instructions. Current best multimodal VLMs are good at predicting semantic actions in the form of *what* is happening at a certain point in a video (Maaz et al. (2023); Huang et al. (2024)), interpreting what objects are in a scene (Achiam et al., 2023) and natively support free-form language conversations for conditioning. However, they are not good at directly predicting *low-level* actions (in the future) of the form of hand-object trajectories. At the same time, recovering low-level interactions in videos, like hand meshes (Pavlakos et al., 2024), object meshes (Fan et al., 2024), and regions of interactions (Shan et al., 2020b; Goyal et al., 2022) have independently become very reliable in recent years. Our key insight is to fine-tune a pre-trained VLM with auto-regressive trajectory predictions of human hand positions, given a few seconds of context video and a language description of the task.

Our approach *HandsOnVLM* casts hand trajectory prediction as an auto-regressive next token prediction conditioned on fused video and language tokens. We develop *HandsOnVLM* as an interactive chat assistant that we can query with informal instructions of the form, "Where should my hand move if I want to open the refrigerator?" and a video (or an image) of a scene, and obtain outputs of the form, "To open the refrigerator, the predicted hand trajectory is $<HAND>$ ,.... $<HAND>$ " The *HandsOnVLM* model first converts the RGB video context to visual tokens and fuses them with the language tokens through slow-fast pooling (Huang et al., 2024) for capturing temporal information from the context video at a fine resolution. We extend the vocabulary to add a new $<HAND>$ token, and output a sequence of text and hand tokens. We finally have a trajectory decoder to convert the hand tokens to a sequence of positions of the left and right hands over the prediction horizon, projected on the image frame.

In summary, our paper has the following contributions:

- We develop *HandsOnVLM*, a novel VLM that can generate textual responses and produce future hand trajectories through conversations by expanding the original vocabulary with hand tokens and having iterative position encodings for auto-regressive predictions during inference.

- We extend existing traditional hand prediction tasks to new tasks, including Vanilla Hand Prediction (VHP) and Reasoning-based Hand Prediction (RBHP), to predict hand trajectories from ego-centric human videos conditioned on language queries of different forms.

- We develop benchmarks for evaluating progress on the VHP and RBHP tasks which we open-source to the community, in addition to our trained models on the respective benchmarks.

Our results on diverse real-world datasets of human videos and zero-shot evaluations on completely unseen datasets demonstrate strong generalization and reasoning capabilities of *HandsOnVLM* for hand trajectory prediction given colloquial language instructions. Furthermore, the model outperforms most baselines on the Reasoning-based Hand Prediction (RBHP) task, showcasing its capability to reason and leverage world knowledge of VLMs.

## 2 Related Work

We discuss prior works on human motion reconstruction and forecasting, developments in multimodal large language models and action understanding from human videos.

### 2.1 Human Motion Prediction

Several prior works have attempted to recover hand meshes and full body meshes from human videos (Rong et al., 2020; Pavlakos et al., 2024). Going beyond reconstruction, other works have also investigated forecasting motions of humans in the future. Early works used RNNs (Bütepage et al., 2017; 2018; Honda et al., 2020) for anticipating future human poses, and recent approaches include Transformer architectures for more diverse and plausible future predictions Ding et al. (2023). More directly related to our work, some approaches predict egocentric hand-trajectories in the form of 2D waypoints (Liu et al., 2020), and others also predict object affordances jointly with hand trajectories (Liu et al., 2022; Bharadhwaj et al., 2024b). Some predict hand trajectories in a 3D space conditioned on a few RGB observations from an egocentric view (Bao et al., 2023). Architectures for such egocentric predictions have ranged from transformers (Liu et al., 2022; Bao et al., 2023) to diffusion models (Ma et al., 2024b;a) trained specifically for this prediction task. Our work extends this line of low-level egocentric trajectory prediction by enabling reasoning capabilities through augmentation and joint training with a pre-trained VLM.

### 2.2 Multimodal Large Language Models

Our work is enabled by developments in multimodal Large Language Models that augment vision and language reasoning in a unified model. Such models like LLaVA (Liu et al., 2024) and Video-ChatGPT (Maaz et al., 2023) have enabled large-scale video understanding and localization of temporal events (semantic actions) in videos (Huang et al., 2024). Adjacently, other works have sought to make the inputs to the VLMs more flexible and informal through automatic segmentations of language instructions (Lai et al., 2024; Yang et al., 2023) and visual grounding allowing flexibility to process both image and region inputs (Rasheed et al., 2024). Recent works have extended the capabilities of VLMs to diverse domains including robotic navigation (Zhang et al., 2024a), robotic manipulation (Kim et al., 2024; Zitkovich et al., 2023), spatial reasoning Cheng et al. (2024), and reasoning about 3D human poses from images and text Feng et al. (2024). While these approaches are orthogonal to our task of egocentric hand trajectory prediction, they serve as evidence of the potential of VLMs for downstream applications.

### 2.3 Action Recognition and Prediction from Videos

Understanding actions in the form of what is happening in a video segment has a long history in computer vision (Sigurdsson et al., 2017; Liu et al., 2021; Kovashka & Grauman, 2010; Feichtenhofer et al., 2019).

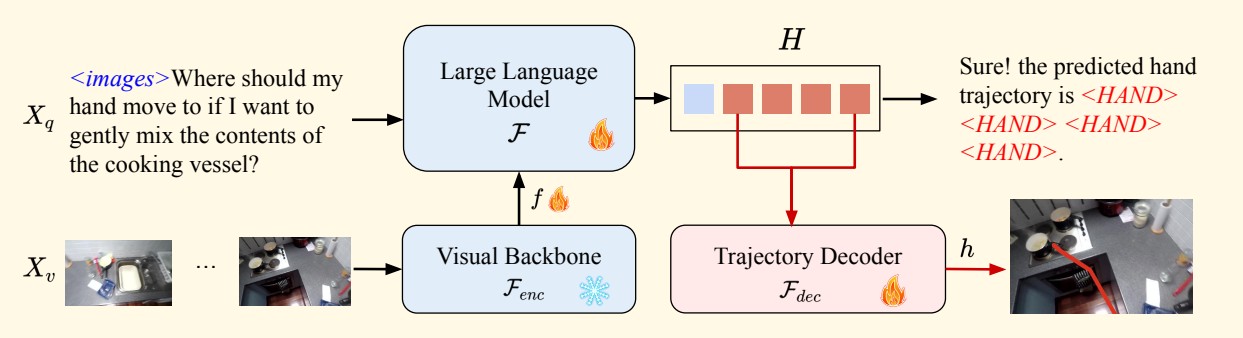

Figure 2: Overview of the *HandsOnVLM* architecture, where 🔥 and ❄ denote trainable and frozen modules separately. *HandsOnVLM* casts hand trajectory prediction as an auto-regressive next token prediction conditioned on fused video and language tokens. The architecture augments a pre-trained VLM with an additional hand token in the vocabulary. We use ⬜ and 🟥 to represent text and $<HAND>$ tokens respectively.

Several benchmarks and datasets containing human videos and action labels for tasks have also been proposed for related problems (Grauman et al., 2022; Caba Heilbron et al., 2015; Goyal et al., 2017; Xiong et al., 2023). Our work leverages such datasets and goes beyond *recognition* of actions in videos to *prediction* of low-level actions in the future by first reasoning about future high-level actions through a VLM. As such our work can have potential applications in robotics for learning motion from web videos for manipulation by complementing prior works in this space (Bharadhwaj et al., 2024a; Bahl et al., 2023; Nair et al., 2022; Bharadhwaj et al., 2025).

## 3 Approach

*HandsOnVLM* is a video-based VLM with the capability of predicting future hand trajectories given a video context and language instructions. There are three key components of *HandsOnVLM*'s architecture: (1) SlowFast tokens to capture temporal information at fine temporal resolution, (2) hand representation using an augmented vocabulary of $<HAND>$ token, and (3) iterative hand decoding to enable auto-regressive trajectory training and inference. In the training stage, we fine-tune a pre-trained VLM by combining next-token prediction loss and trajectory loss.

### 3.1 Architecture

We show an overview of the *HandsOnVLM* model architecture in Fig 2. *HandsOnVLM* takes a sequence of $T$ frames $X_v$ and a language instruction $X_q$ as input and predicts future hand trajectories $\mathcal{H} = \{h_{T+i}\}_1^N$, where $N$ is the future horizon. At each future time step $T + i$, the future hand location $h_{T+i}$ consists of the 2D location of the center of the left and right hands projected to the last observation frame $X_v[-1]$. The key components of the architecture include a visual backbone $\mathcal{F}_{enc}$, a vision-to-language projection layer $f$, a Large Language Model(LLM) $\mathcal{F}$ and a trajectory decoder $\mathcal{F}_{dec}$.

**SlowFast Token Compression.** To obtain a capable video-conditioned VLM we need to be able to interpret temporal information at a fine resolution. Following Huang et al. (2024), given $X_v$, we embed them into $T \times M$ visual tokens using a visual backbone, where $M$ is the number of tokens in each frame. Then we apply slow-fast pooling to get $T + M$ visual tokens. In the fast path, we average all the tokens within each frame to get $T$ tokens overall. We also uniformly select $s$ frames among all $T$ frames and perform $s \times s$ spatial average pooling to get $M$ slow frames in total. These slow tokens will help preserve spatial information during the encoding process. Then we embed and align $T + M$ visual tokens to the language space through a vision-to-language projector $f(\cdot)$.

**Hand as Embedding.** To represent hand in the language space, we extend the existing vocabulary with a new $<HAND>$ token. However, a typical embedding layer would encode each $<HAND>$ token identically,

resulting in individual $<HAND>$ token being indistinguishable from one another. To overcome this limitation, we embed ground truth hand positions into the $<HAND>$ tokens during the tokenization process. We feed them into the Large Language Model backbone and get the embedding of the last layer $H$, where $H = \mathcal{F}(X_q, f(\mathcal{F}_{enc}(X_v)))$.

**Iterative Hand Decoding.** For $i$-th token in the sequence, let $H_i$ be the last-layer embedding of this token from the Large Language Model. *HandsOnVLM* decode it to predict the $(i+1)$-th token as LLMs do. When $(i+1)$-th token is a $<HAND>$ token, we input $H_i$ into a hand trajectory decoder $\mathcal{F}_{dec}$ to predict the hand position of the $(i+1)$-th token $h_{i+1} = \mathcal{F}_{dec}(H_i)$. During inference, this decoded position is then encoded into the corresponding $<HAND>$ token embedding for following prediction rounds. In this way, we ensure that each subsequent prediction is conditioned on all previously predicted hand positions, maintaining temporal consistency and spatial awareness throughout the inference process and mitigating compounding errors.

## 3.2 Training Objectives

The model is trained end-to-end using a text generation loss $\mathcal{L}_{\text{txt}}$ and a hand trajectory prediction loss $\mathcal{L}_{\text{hand}}$. The overall objective $\mathcal{L}$ is the weighted sum of both losses, determined by $\lambda_{\text{txt}}$ and $\lambda_{\text{hand}}$:

$$\mathcal{L} = \lambda_{\text{txt}}\mathcal{L}_{\text{txt}} + \lambda_{\text{hand}}\mathcal{L}_{\text{hand}} \tag{1}$$

Specifically, $\mathcal{L}_{\text{txt}}$ is the auto-regressive cross-entropy loss for text generation, and $\mathcal{L}_{\text{hand}}$ is the hand prediction loss, which encourages the model to generate high-quality hand trajectories as well. Following Liu et al. (2022), we employ a reconstruction loss over future timesteps and a KL-Divergence Regularization loss as $\mathcal{L}_{\text{hand}}$:

$$\mathcal{L}_{\text{hand}} = \sum_{t=1}^{N} \mathcal{L}_{\text{recon}} \left(h_{T+t}, \hat{h}_{T+t}\right) + \mathcal{L}_{kl}\left(\mu_h, \sigma_h\right). \tag{2}$$

We employ CVAE (Sohn et al., 2015) as the hand trajectory decoder in this work (although the method is not tied to it). Thus, $\mathcal{L}_{\text{recon}}$ is the MSE loss over valid hand positions, and $\mu_h$, $\sigma_h$ here are the mean and the standard deviation that regularizes the latent z-space to be close to the normal distribution.

# 4 Reasoning and Predicting Hand Trajectories

In this section, we introduce two tasks: the Vanilla Hand Prediction (VHP) task, which extends the classic hand motion prediction (Liu et al., 2022), and the proposed Reasoning-based Hand Prediction (RBHP) task. Finally, we describe a two-step annotation-generating pipeline to build the corresponding RBHP dataset.

## 4.1 Vanilla Hand Prediction Task

In this task, explicit action narration is required to predict the next hand motion. Here explicit means the action narration directly specifies the action and the target object without ambiguity, such as "cut the paper" or "open the microwave". We choose Epic-Kitchen (Damen et al., 2018; 2022), H2O (Kwon et al., 2021) and FPHA (Garcia-Hernando et al., 2018) as datasets for this task. To generate the hand labels for all the datasets, following (Liu et al., 2022), we first run an off-the-shelf active hand-object detector (Shan et al., 2020a) to get the hand bounding box in each frame. To get the ground truth of each future hand trajectory, we first compute pairwise homographies by matching SURF (Bay, 2006) features of masked regions through RANSAC and project each future hand position into the last observation frame. Then, we apply cubic Hermite spline interpolation to smooth the projected trajectories and fill any missing points. Finally, we filter the resulting trajectories with multiple criteria, including confidence thresholds, highest-score detection selection, feature matching thresholds, trajectory completeness checks, and boundary constraints.

To reformat these datasets for visual question answering, we structure them in a question-answer format using the following template:

*"USER:<images>, can you give me the future hand trajectory for {explicit action narration}? ASSISTANT: Sure, it is<HAND>...<HAND>.",*

where *<images>*represents a placeholder of visual tokens of the input frames. Note that the action is optional because we can also generate general templates without specifying the action, and in this case the task reduces to that in prior works (Liu et al., 2022; Bao et al., 2023; Ma et al., 2024b).

## 4.2 Reasoning-based Hand Prediction Task

In addition to the Vanilla Hand Prediction Task, we introduce the Reasoning-based Hand Prediction (RBHP) task. Instead of utilizing explicit instructions to directly predict the hand motion, here the system is required to reason about it with implicit instructions. We define implicit instructions as colloquial language instructions that provide sufficient information for inferring the intended human hand action through reasoning, without explicitly naming the target object or action.

To construct a dataset for this task, we implement a two-step annotation-generating pipeline (Fig. 3) powered by GPT-4 (Achiam et al., 2023). This pipeline extracts implicit instructions from the Epic-Kitchens-100 dataset (Damen et al., 2022). Prompt templates for these two steps are provided in the Appendix.

**Classic Hand Trajectory Dataset**

Action: Get the dough

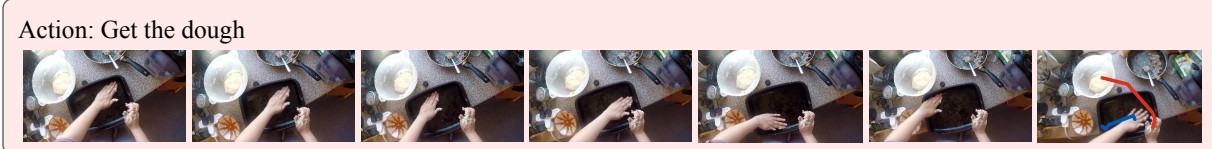

**GPT-4 Action-aware Image Description**

In the image, there is a countertop with various kitchen items. A hite mixing bowl containing dough is positioned on the left, while a black baking tray sits in the center. The person's hand, covered in dough, is reaching towards the tray, suggesting they are about to transfer the dough. To the right, there is a frying pan with some cooked meat, and a blender is visible in the background. The overall scene depicts a busy kitchen environment, focused on food preparation.

**Implicit Action Generation**

*<images>* What is the recommended hand movement for accessing the mixture in the white container?

**Visual Question-Answering Conversion**

To accessing the mixture in the white container, the recommended hand trajectory is: *<HAND><HAND><HAND><HAND>*.

Figure 3: Illustration of the annotation pipeline for the RBHP task. By using GPT-4 on human video datasets we extract implicit language instructions for visual question-answering. The red and blue lines respectively show trajectories for the right and left hands.

**Action-aware Image Description.** To get the implicit instructions, the first step is to generate a detailed description of the scene including all the objects in the foreground. We prompt GPT-4 with the ground truth action to capture action-related information, such as the physical properties of the target object or the spatial relationship with other objects.

**Implicit Action Generation.** Using the action-aware description of the scene, we are able to generate the implicit instructions using GPT-4 in a text template as follows:

*"USER:<images>, can you give me the future hand trajectory for {action implicit description}? ASSISTANT: Sure, it is <HAND>..<HAND>.".*

We choose Epic-Kitchen and Ego4D (Grauman et al., 2022) as datasets for this task. Through the annotation-generating pipeline, we generate 7.5k question-answering pairs from Epic-Kitchen, and 8k pairs from Ego4D for zero-shot evaluation.

| | | On Validation Split | | | | | | Zero-shot | | | | | |
|---|---|---|---|---|---|---|---|---|---|---|---|---|---|
| | | EK55 | | | EK100 | | | H2O | | | FPHA | | |
| Approach | BBox Input | ADE ↓ | FDE ↓ | WDE ↓ | ADE ↓ | FDE ↓ | WDE ↓ | ADE ↓ | FDE ↓ | WDE ↓ | ADE ↓ | FDE ↓ | WDE ↓ |
| KF | ✓ | 0.392 | 0.386 | 0.199 | 0.317 | 0.318 | 0.168 | - | - | - | - | - | - |
| OCT | ✓ | 0.216 | 0.199 | 0.105 | 0.209 | 0.187 | 0.102 | - | - | - | - | - | - |
| OCT-global | | 0.232 | 0.218 | 0.115 | 0.216 | 0.193 | 0.105 | - | - | - | - | - | - |
| LLaVA-Pixel2Seq | | 0.156 | 0.139 | 0.076 | 0.254 | 0.224 | 0.124 | 0.150 | 0.121 | 0.032 | 0.214 | 0.189 | 0.043 |
| LLaVA-Traj | | **0.126** | 0.142 | 0.073 | 0.201 | 0.191 | 0.103 | **0.133** | 0.130 | 0.031 | 0.191 | 0.167 | 0.041 |
| *HandsOnVLM* | | 0.136 | **0.106** | **0.062** | **0.194** | **0.157** | **0.090** | 0.135 | **0.108** | **0.028** | **0.175** | **0.151** | **0.034** |

Table 1: Comparison of VHP task with different baselines. We reported the performance on the validation split of Epic-Kitchen dataset. For the RBHP baselines, we also evaluate them on two unseen datasets, H2O and FPHA.

# 5 Experiments

We perform experiments for both the proposed tasks in order to answer the following research questions:

- How plausible are the hand trajectories produced by *HandsOnVLM*?

- Does *HandsOnVLM* exhibit reasoning abilities for implicit language queries?

- Does *HandsOnVLM* generalize zero-shot to unseen scenes from new datasets?

## 5.1 Experiment Details

**Architecture.** Following LITA's architecture, We use CLIP-L-14 (Radford et al., 2021) as the visual encoder and Vicuna (Chiang et al., 2023) as the LLM module. We adapt the vision-language projector from LLaVA (Liu et al., 2024) and have a CVAE (Sohn et al., 2015) as trajectory decoder.

**Datasets.** For VHP and RBHP datasets, we sample 10 frames and predict the hand position in next 4 frames at FPS = 4. In addition to our proposed datasets, *HandsOnVLM*[†] are also trained on a few additional datasets for five different tasks, namely ActivityNet-Captions (Krishna et al., 2017)and YouCook2 (Zhou et al., 2018) for dense video captioning and event localization, NExT-QA  (Xiao et al., 2021) for video question answering, LLaVA-150K (Liu et al., 2024) for image instruction tuning, ActivityNet-RTL (Huang et al., 2024) for reasoning temporal localization. We co-train with these additional tasks to help with visual understanding and reasoning, and this is enabled by the flexible modeling of *HandsOnVLM* that allows training on generic QA datasets.

**Implementation Details.** For *HandsOnVLM* and other VLM-based baselines, in each epoch we select 24K samples from the Epic-Kitchens-100 VHP dataset. For *HandsOnVLM*[†], in each epoch we randomly select 6K samples in Epic-Kitchens-100 VHP dataset, 6K in Epic-Kitchens-100 RBHP dataset and another 12K that are uniformly distributed among all other 5 tasks. We use a batch size of 128, a learning rate of 2e-5 and train for 40 epochs. The total wall-clock time for training is around 18 hours for the 7B models while using 8 H100 GPUs. The LLM and vision-language projector are initialized with the LLaVA-1.3 pre-trained weights. During training, we freeze the visual backbone and fully fine-tune other modules.

## 5.2 Metrics and Baselines

Following previous works (Liu et al., 2022; Ma et al., 2024b) we use Average Displacement Error (ADE), Final Displacement Error (FDE) and Weighted Displacement Error (WDE) as metrics to evaluate VHP and RBHP tasks.

**Vanilla Hand Prediction.** For the VHP task, we choose Kalman Filter(KF) and Object-centric Transformer(OCT) (Liu et al., 2022) as the baselines. Since OCT still requires the bounding box feature of the hand and object as input, to get a fairer comparison with other end-to-end methods, we implement a version without the requirement of the bounding box, which we call OCT-global.

| Approach | RBHP (Epic-K) | | | RBHP (Ego4D) | | |
|---|---|---|---|---|---|---|
| | ADE ↓ | FDE ↓ | WDE ↓ | ADE ↓ | FDE ↓ | WDE ↓ |
| Kling 1.5 | 0.31 | 0.35 | 0.19 | 0.27 | 0.41 | 0.18 |
| LumaLabs | 0.29 | 0.37 | 0.18 | **0.21** | 0.28 | 0.13 |
| LLaVA-P2S | 0.27 | 0.24 | 0.13 | 0.31 | 0.28 | 0.14 |
| LLaVA-T | 0.19 | 0.18 | 0.10 | 0.38 | 0.35 | 0.17 |
| *HandsOnVLM* | 0.19 | 0.16 | 0.09 | 0.22 | 0.19 | 0.10 |
| *HandsOnVLM*[†] | **0.18** | **0.15** | **0.08** | 0.22 | **0.18** | **0.09** |

Table 2: Comparison of *HandsOnVLM* on the RBHP task with different baselines. †means fine-tuned on the RBHP dataset.

**Reasoning-based Hand Prediction.** To evaluate *HandsOnVLM*'s performance on the RBHP task, we perform baseline comparisons with several VLM-based methods. We describe these basleines below:

- **LLaVA-Traj.** Note that the hand trajectories are a sequence of pixel positions, we can represent them in text directly. In this case, we can directly fine-tune the LLaVA without any modification.

- **LLaVA-Pixel2Seq.** An alternative approach to representing hand positions involves quantizing the image into discrete spatial bins (Chen et al., 2021), each corresponding to a unique token. We can extend the existing vocabulary with those discrete tokens.

- **Language-conditioned Image-to-Video Models.** We also compare our model to baselines of the language-conditioned image-to-video generation followed by hand-tracking. We use commercial state-of-the-art language-conditioned image-to-video systems such as LumaLabs (LumaLabs, 2024), Kling 1.5 (KlingAI, 2024) and generate videos conditioned on the last observation frame and the language description. Following the hand label generation process in Sec. 4.1, we track and extract the hand trajectories of the generated video.

## 5.3 Comparisons with Baselines

We evaluate *HandsOnVLM* on both the VHP task and the proposed RBHP task and report the results and comparisons with baselines in Table 1 and Table 2 respectively. All models except *HandsOnVLM*[†] are trained on VHP datasets. *HandsOnVLM*[†] is trained on all available datasets (Data Combo 5 in Table 3).

**VHP Task.** We evaluate all the baselines on the VHP datasets as described in section 5.1. Here, the FPHA and H2O datasets serve as unseen datasets to test zero-shot generalization capabilities. Among all the VHP datasets, *HandsOnVLM* outperforms both the task-specific methods as well as the VLM-based methods, which demonstrates its strong ability to produce plausible trajectories corresponding to how a real human hand would move given explicit instructions. We also find that *HandsOnVLM* can generalize to completely unseen scenes (for example scenes from H2O and FPHA datasets), which demonstrates it can effectively leverage the world knowledge of the pre-trained VLM.

**RBHP Task.** For evaluations on the RBHP task shown in Table 2, *HandsOnVLM* achieves state-of-the-art performance in all three metrics. This suggests that *HandsOnVLM* is able to reason based on implicit cues of the scene and be applied to complicated scenarios involving everyday natural language conversations. However, we observe that LumaLabs (LumaLabs, 2024) achieves the smallest ADE in the Ego4D RBHP benchmark but relatively higher FDE and WDE. This may be because the commercial text-conditioned image-to-video generation models have realistic video generation capabilities but cannot understand reasoning-based language prompts which is necessary for generating plausible videos maintaining temporal consistency. Since the training dataset compositions of these video models are not disclosed, there may also be some data leakage issues of the evaluation datasets in this paper being a part of their training corpora.

| Data Combos | Epic-Kitchen | | LITA data | RBHP data | ADE↓ | FDE↓ | WDE↓ |
|---|---|---|---|---|---|---|---|
| | 55 | 100 | | | | | |
| 1 | ✓ | | | | 0.206 | 0.195 | 0.101 |
| 2 | ✓ | ✓ | | | 0.197 | 0.165 | 0.094 |
| 3 | ✓ | ✓ | ✓ | | 0.199 | 0.163 | 0.094 |
| 4 | ✓ | ✓ | ✓ | ✓ | **0.187** | **0.156** | **0.089** |

Table 3: Analysis of the impact of training data on the performance of *HandsOnVLM*. We can see that performance increases with additional data of VHP (first two rows), even with datasets of other tasks (third row), but the highest gains come from the proposed RBHP dataset (last rows).

| Num of Generations | ADE↓ | FDE↓ | WDE↓ |
|---|---|---|---|
| 1 | 0.187 | 0.156 | 0.089 |
| 4 | 0.184 | 0.152 | 0.087 |
| 8 | **0.182** | 0.151 | **0.086** |
| 16 | **0.182** | **0.150** | **0.086** |

Table 4: Analysis of test-time computations for *HandsOnVLM* in the form of stochastic decoding with self-consistency (Wang et al., 2023).

## 5.4 Ablation Study

In this section, we conduct a broad study of the different components of our model. All experiments in this section are evaluated on the RBHP task.

**Effects of Different Sources of Dataset.** In Table 3, we show the contribution of each type of dataset to the performance of *HandsOnVLM*. LITA dataset denotes the different datasets for 5 additional tasks (Huang et al., 2024) described in Section 5.1 ranging from dense video captioning to reasoning about temporal localization. While increasing the scale of the VHP dataset (first two rows) can bring some improvement, we find that fine-tuning with the reasoning dataset (last two rows) can significantly boost the performance, even when fine-tuning with tasks that are not directly related to hand trajectory prediction. This demonstrates that *HandsOnVLM* can leverage world knowledge learned by other tasks to reason about predicting plausible hand trajectories.

**Test-time Computation.** Recent works (Snell et al., 2024; OpenAI, 2024) have shown that using more test-time computation is a critical step for LLMs to improve their performance, especially on reasoning tasks. Motivated by these, we also investigate if such properties can enhance the performance of *HandsOnVLM* predictions. We report the performance using different numbers of generations during the stochastic decoding with self-consistency (Wang et al., 2023) in Table 4. The main idea is to sample a diverse set of reasoning paths instead of just one and then select the most consistent output through marginalization. To obtain the self-consistency result in our context, we generate multiple answers for each inquiry and then average the predicted hand trajectory. We find that increasing the test-time computation in this form can robustly improve the performance of *HandsOnVLM* as seen by the lower metrics from top to bottom in Table 4 .

**Effect of Different Observation Frames.** In Table 7, we investigate the performance of our approach and the baselines when conditioned on only one observation frame instead of an observation video. Here we have four comparisons: OCT-last-im, OCT-global-last-im, *HandsOnVLM*-last-im, *HandsOnVLM*†-last-im, which respectively correspond to versions of our baselines in Sec. 5.2 but are only conditioned on the last frame of the input video context. We find that the results in this evaluation scenario are comparable to the setting where the context is a video, indicating that HandsOnVLM can be flexibly conditioned on just one image when a video context is not available.

**Effect of Fine-tuning Visual Backbone.** We follow the standard practice of training the Vision-language Model, freezing the visual backbone during training. Many previous works Liu et al. (2024); Feng et al. (2024); Huang et al. (2024) have demonstrated the effectiveness of this method. Due to computational resource limitations, we conduct the experiments for thsi ablation study on a relatively small scale, which

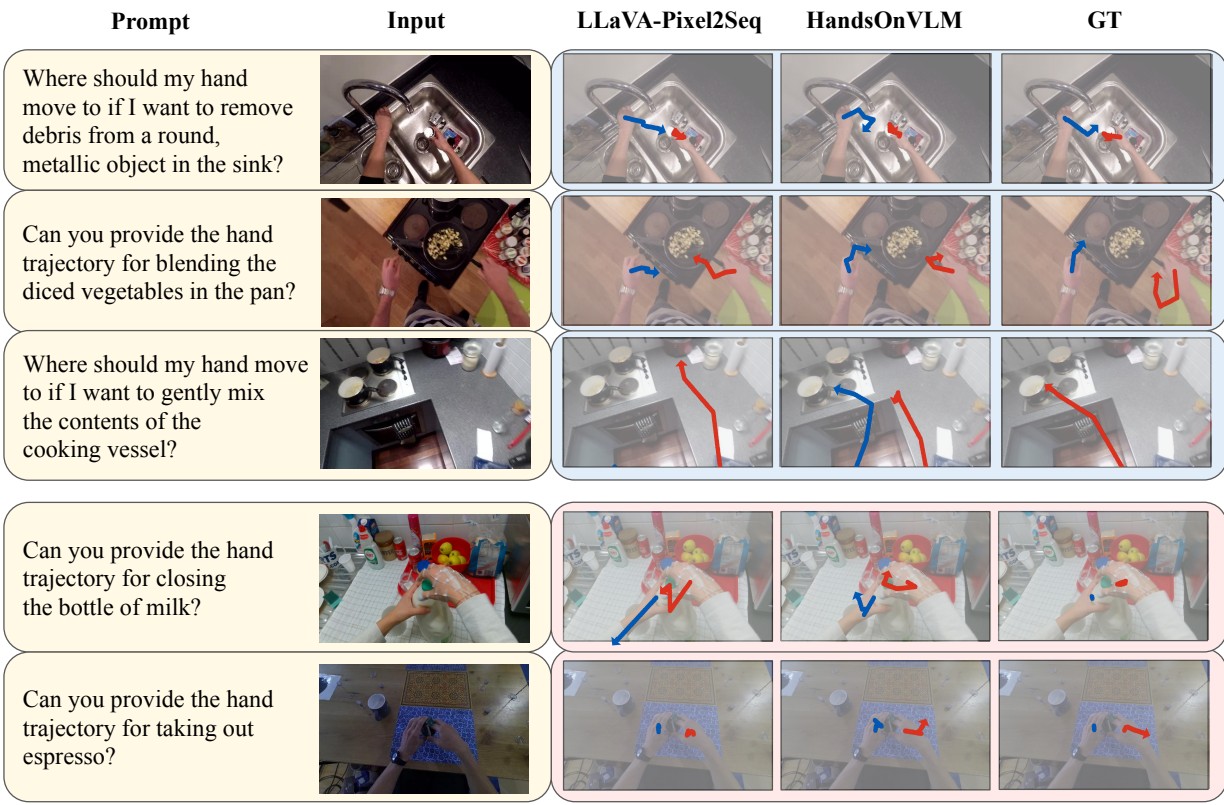

Figure 4: Qualitative results for different samples from the validation split of our RBHP dataset (top in blue) and zero-shot evaluations on completely unseen datasets FPHA and H2O (bottom in pink). The left-hand trajectory is visualized in blue and the right-hand trajectory is in red. The arrows denote the direction of each trajectory. GT trajectories are provided for reference.

| Task | Methods | ADE↓ | FDE↓ | WDE↓ |
|------|---------|------|------|------|
| VHP | OCT | 0.209 | 0.187 | 0.102 |
| | OCT-last-im | 0.213 | 0.191 | 0.104 |
| | OCT-global | 0.216 | 0.193 | 0.105 |
| | OCT-global-last-im | 0.212 | 0.189 | 0.103 |
| | *HandsOnVLM* | **0.194** | **0.157** | **0.090** |
| | *HandsOnVLM*-last-im | 0.197 | 0.165 | 0.094 |
| RBHP | *HandsOnVLM* | 0.197 | 0.165 | 0.094 |
| | *HandsOnVLM*-last-im | 0.197 | 0.163 | 0.093 |
| | *HandsOnVLM*† | **0.187** | 0.156 | 0.089 |
| | *HandsOnVLM*†-last-im | **0.187** | **0.155** | **0.088** |

Table 5: Analysis of the # of observation frames during inference.

is named VHP-Small in Table 6. We find that fine-tuning the visual backbone degrades performance. This observation is aligned with some previous works in the VLM space Karamcheti et al. (2024).

**Effect of Slow-fast Tokens.** Many previous works have suggested that slow-fast tokens are beneficial for video understanding. In workFeichtenhofer et al. (2019), the authors have demonstrated the contribution of slow-fast tokens to video action classification and detection tasks. A recent work Huang et al. (2024) implemented a similar design in video question answering problems and achieved state-of-the-art performance

| Task | Methods | ADE↓ | FDE↓ | WDE↓ |
|---|---|---|---|---|
| VHP-Small | Fine-fine Visual Backbone | 0.192 | 0.178 | 0.100 |
| | Frozen Visual Backbone | **0.174** | **0.154** | **0.089** |

Table 6: Analysis of fine-tuning visual backbone during training.

| Task | Methods | ADE↓ | FDE↓ | WDE↓ |
|---|---|---|---|---|
| | Fast Token Only | 0.316 | 0.236 | 0.144 |
| VHP-Small | Slow Token Only | 0.200 | 0.174 | 0.100 |
| | Slow-fast Tokens | **0.174** | **0.154** | **0.089** |

Table 7: Analysis of slow-fast tokens during training.

in video understanding and answering tasks. We find that among all the options, slow-fast tokens work the best, proving the effectiveness of our design choice.

## 5.5 Qualitative Results

In Fig. 4 we show qualitative results for *HandsOnVLM* and the strongest baseline LLaVA-Pixel2Seq. The section above the horizontal line shows visualization from the validation split of RHBP datasets, while the section below the line shows *zero-shot* results on scenarios from completely unseen datasets.

In the second row, we observe that *HandsOnVLM* generates a trajectory where the left hand stably holds the pan while the right hand performs the blending action. In contrast, LLaVA-Pixel2Seq fails to correctly predict the act of holding the pan. The third row results demonstrates *HandsOnVLM*'s ability to reason about multimodal solutions for the same task. While the ground truth shows that the right hand moving the pot, *HandsOnVLM* chooses to use the left hand to execute the same action, illustrating its multimodal reasoning capability.

## 5.6 Human Evaluation

Going beyond automated metrics, to determine plausibility of the generated hand trajectories for various scenarios, we also perform human evaluations. We create a simple user interface that contains the context video along with a language description of the task, and present two separate videos corresponding to predicted hand trajectories (for two different methods). We ask the users to pick which prediction appears more plausible given the context video and task description. We randomize the relative orders across different instances, and after the user clicks on a prediction, the page refreshes showing the the next scenario. Overall, we perform this study with 100 examples each for the VHP and RBHP settings, and recruit 20 participants for the study. Results in Table 8 show that the predictions from *HandsOnVLM*[†] (the model fine-tuned on reasoning tasks) is more plausible for both VHP and RBHP evaluations, suggesting that model is able to effectively leverage world knowledge from other reasoning tasks to reason about low-level hand-object interaction predictions in diverse scenarios.

## 6 Conclusion

In this work, we propose *HandsOnVLM*, a novel video-based VLM to predict future hand motion from ego-centric video contexts. We also propose different prediction tasks, including Vanilla Hand Prediction

| Method | VHP | RBHP |
|---|---|---|
| *HandsOnVLM* | $30 \pm 3\%$ | $28 \pm 5\%$ |
| *HandsOnVLM*[†] | $70 \pm 3\%$ | $72 \pm 5\%$ |

Table 8: Human study showing the % mean and SE of trials where participants consider hand trajectory predictions from one method more plausible than the other.

(VHP) and Reasoning-based Hand Prediction (RBHP) to benchmark low-level trajectory prediction and reasoning. We demonstrate the effectiveness of our approach through extensive evaluations in diverse real-world scenarios. We believe this research represents a promising initial step towards integrating egocentric hand-object trajectory predictions with the powerful reasoning capabilities of VLMs. As the ground-truth hand reconstruction from videos gets better due to wide community interest, the capabilities of our prediction model can further improve in scenarios like occlusion, and be practically deployable for in-context predictions in wearable devices that are becoming ubiquitous. An exciting direction of future work would be to adapt our model for long-horizon predictions for activities like "making coffee" which would consist of several steps and require reasoning over an extended period. Since video clips on the web have significant camera motion over time, a viable strategy for this could be chaining the model sequentially for different sub-tasks.

## Acknowledgements

We thank Gaurav Parmar and Jinkun Cao for feedback on the paper, and thank Yufei Ye, Ruihan Yang, Unnat Jain, Mohan Kumar Srirama, Shubham Tulsiani, and many others from CMU and UCSD for helpful discussions. This work used Bridges-2 at Pittsburgh Supercomputing Center from the Advanced Cyberinfrastructure Coordination Ecosystem: Services & Support (ACCESS) program, which is supported by National Science Foundation grants 2138259, 2138286, 2138307, 2137603, and 2138296.

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

# A   Appendix

Here we provide additional details of the model implementation, dataset curation, and more qualitative results.

## A.1   Dataset Details

**Statistics.** Table 9 shows the statistics of all datasets used in our tasks. Note that H2O, FPHA and Ego4D are only used for zero-shot evaluation so there are no training samples.

| Task | Dataset | Training Samples | Validation Samples |
|------|---------|------------------|--------------------|
| VHP | Epic-Kitchen-55 | 8523 | 1894 |
| | Epic-Kitchen-100 | 24148 | 3513 |
| | H2O | - | 503 |
| | FPHA | - | 501 |
| RBHP | Epic-Kitchen-100 | 4018 | 3513 |
| | Ego4D | - | 8673 |

Table 9: Data Statistics of VHP and RBHP task.

## A.2   Pipeline Details

Here we provide the illustrations of the training pipeline and the inference pipeline.

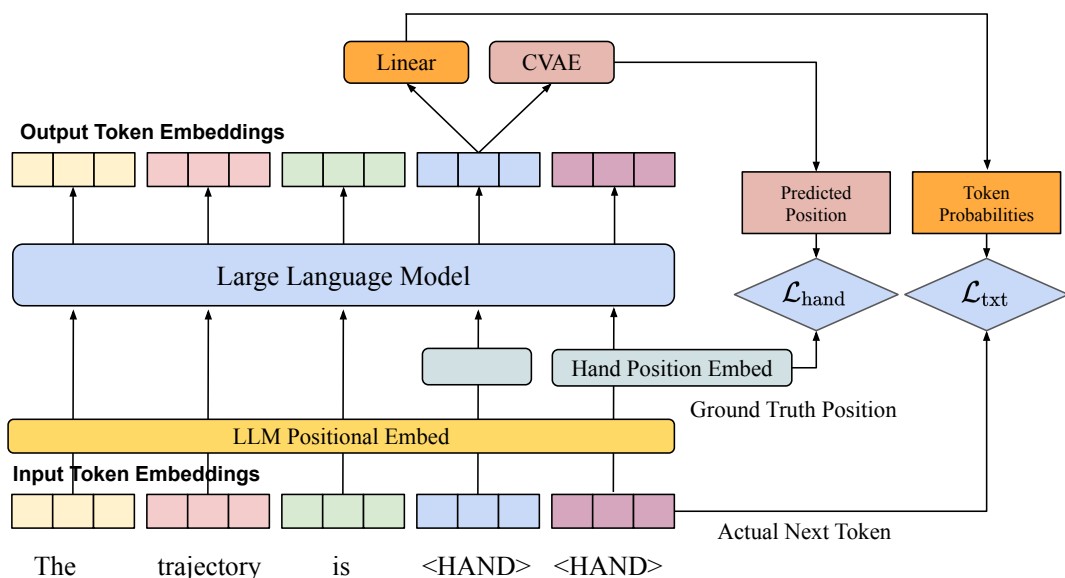

Figure 5: Illustration of training pipeline.

## A.3   Other Ablation Studies

**Scaling Model Improves the Prediction.** To evaluate the scaling ability of our model, we use LLaVA-V1.5-7B and LLaVA-V1.5-13B as the LLM backbone of our model. We refer them as *HandsOnVLM*-7B and *HandsOnVLM*-13B. We show the performance of both models in Fig. 7.

**Zero-shot Chain-of-thought.** We also conduct an ablation study on the zero-shot chain-of-thought (Wei et al., 2022; Kojima et al., 2022) prompting, as shown in Fig. 8. We add "Let's think step by step" in the

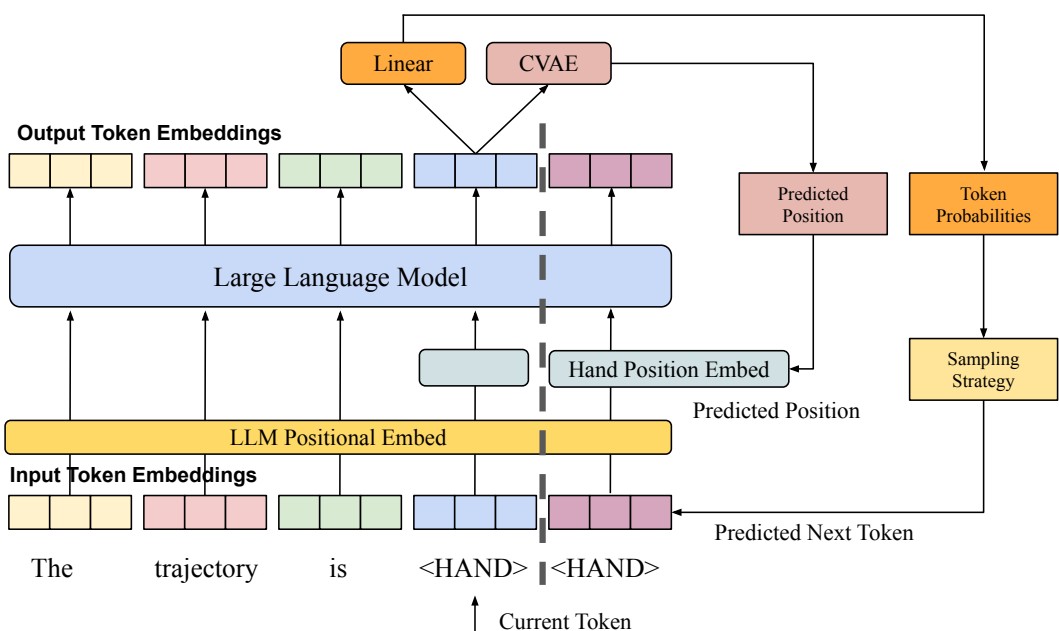

Figure 6: Illustration of inference pipeline.

front of the answer generated in the inference stage. Contrary to our expectations, this approach yielded poorer results. This unexpected outcome may be attributed to the limited diversity of our datasets.

| Approach | ADE↓ | FDE↓ |
|---|---|---|
| *HandsOnVLM*-7B | 0.197 | 0.165 |
| *HandsOnVLM*-13B | **0.183** | **0.149** |

Figure 7: Ablation study on the LLM backbone size. We evaluate them on the RBHP task.

| Reasoning Method | ADE↓ | FDE↓ |
|---|---|---|
| Direct Answer | **0.197** | **0.165** |
| Chain-of-Thought | 0.220 | 0.191 |

Figure 8: Comparison of direct answer and chain-of-thought reasoning methods.

## A.4 More Visualizations

**Failure Cases.** We show some failure cases in Fig. 9. We observe failures when (1) there are someone's hands in the video, (2) the hands are occluded by objects, and (3) the target object in the instruction is not found in the frame.

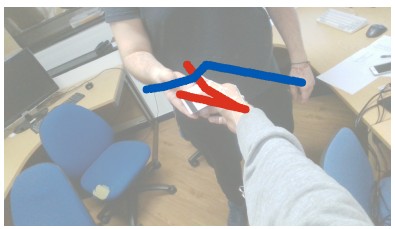 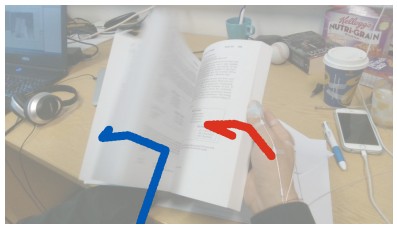 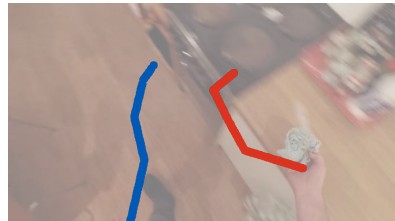

Figure 9: Failure cases of the model: (left) multiple hands in the video, (middle) occlusions, and (right) the target trash can is out of view.

**More Qualitative Results.** We provide more visualizations in Fig. 10.

### A.5 Prompt for VHP and RBHP Dataset Generation

We provide the GPT4 prompts for the RBHP dataset generation pipeline mentioned in Section 4.2 in Table 10 and Table 11.

| GPT4 Prompt for Action-aware Image Description |
|---|
| You are a system generating descriptions for ego-centric human images. Human is doing household activities.

Provided with an image and a action narration of what is happening next, such as "use the scissor", you will describe the main item that you see in the image, giving details but staying concise.

You can describe unambiguously what the item is, its color or relative position if clearly identifiable.
You should also give out a overall description of the scene, the environment where the action is taking place. |

Table 10: GPT4 prompt for action-aware image description.

| GPT4 Prompt for Implicit Action Generation |
|---|
| You are tasked with creating specific, indirect questions and instructions that human could use to identify and interact with objects based on their names or detailed descriptions provided by users.

You will be given an action phrase which the human is going to do next, such as "use the scissor".

Based on the descriptions, you must formulate responses that precisely hint at the action phrase without naming it directly. The aim is to enable the agent to deduce the correct action through these indirect cues, enhancing its ability to understand and execute tasks involving the object.

Please format your generated response as a hand trajectory question, some templates are provided below for reference:
"Where should my hand move to if I want to {implicit description}"
"Can you provide the hand trajectory for {implicit description}?"
"What is the recommended hand movement for {implicit description}?" |

Table 11: GPT4 prompt for implicit action generation.

| Question Templates to Build VHP Datasets. |
|---|
| "Can you provide the hand trajectory?"
"What is the recommended hand movement?"
"What is the future hand trajectory in this video?"
"What is the predicted hand trajectory given current observations?"
"Where should my hand move to if I want to {explicit action}?"
"Can you provide the hand trajectory for {explicit action}?"
"What is the recommended hand movement for {explicit action}?" |

Table 12: Question Templates to build VHP datasets.

| Answer Templates to build VHP and RBHP datasets. |
|---|
| "Sure! Here is the hand trajectory {hand token sequence}."
"Based on the video, the hand trajectory is as follows: {hand token sequence}."
"The predicted hand trajectory is as follows: {hand token sequence}."
"Certainly! The hand trajectory for {action instruction} is as follows: {hand token sequence}."
"To {action instruction}, the recommended hand trajectory is: {hand token sequence}." |

Table 13: Answer Templates to build VHP and RBHP datasets.

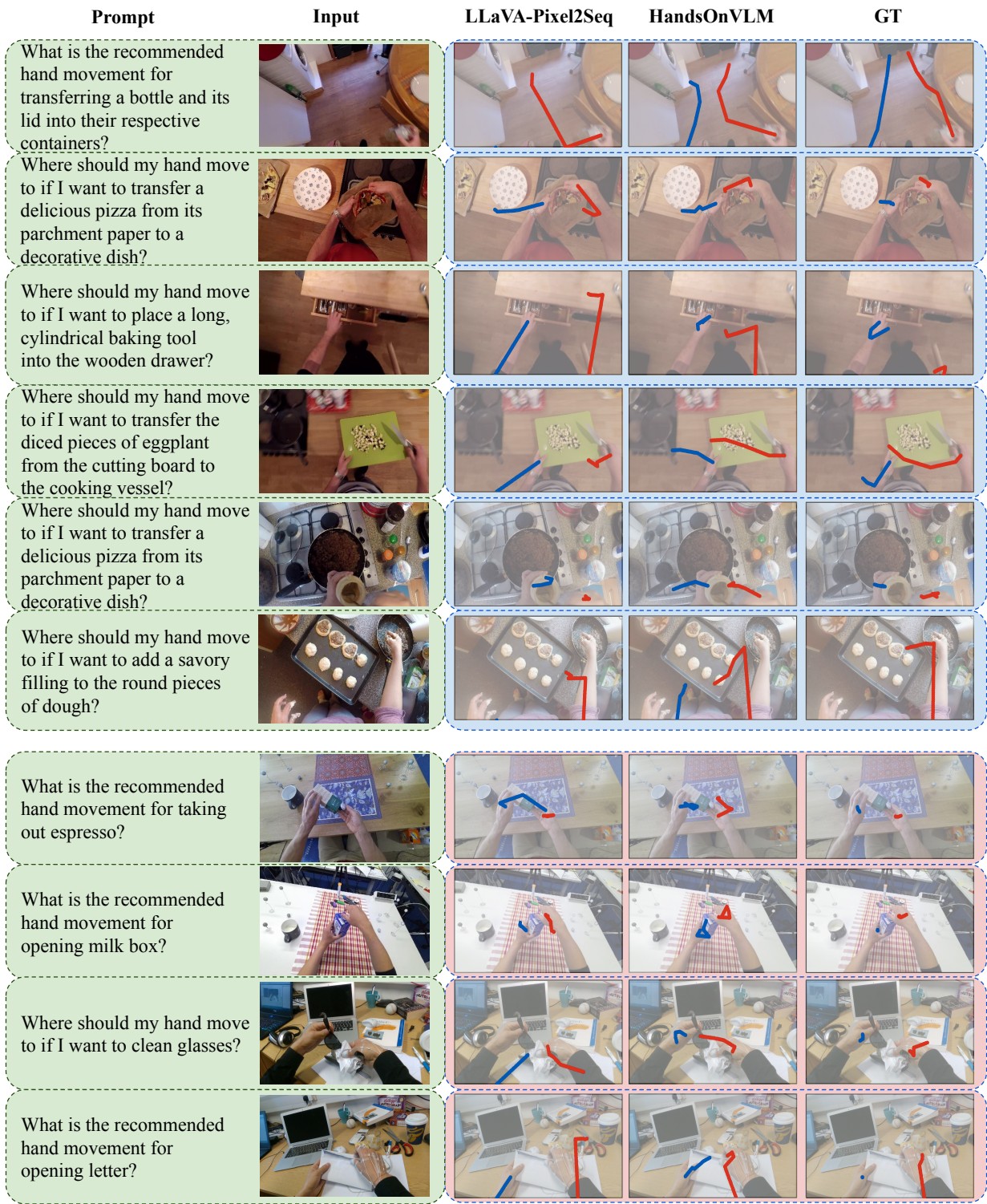

Figure 10: More Qualitative results for different samples from the validation split of our RBHP dataset (top in blue) and zero-shot evaluations on completely unseen datasets FPHA and H2O (bottom in pink). GT trajectories are provided for reference.

