# OpenReview forum: "HandsOnVLM: Vision-Language Models for Hand-Object Interaction Prediction"
_TMLR — Accepted by TMLR_

### Review · Reviewer_EfpJ · 2025-06-19

**Summary Of Contributions:**

This paper proposed a new framework called HandsOnVLM. It leverages the inputs of videos and text prompts to predict the hand position in the next frame. The author also proposed new datasets and tasks.

**Audience:**

Yes

**Broader Impact Concerns:**

None.

**Claims And Evidence:**

No

**Requested Changes:**

Please refer to the **Weaknesses**.

**Strengths And Weaknesses:**

1. A novel framework for hand position prediction.
2. Proposed datasets are good.


**Weaknesses**
1. I didn't see clear difference between the newly extended tasks and the corresponding original tasks. Please state them clearly.
2. The paper writing is somehow bad, making it hard to read.
3. This work is more likely to be a technical work, especially the design of the model framework, which mostly involve the existing techniques.
4. What is the future horizon? I did not see explicit definition in the paper.
5. How did the authors extend the existing vocabulary with a new <HAND> token?
6. Regarding the establishment of the proposed datasets, the authors use some existing models to obtain the 'pseudo-GT', how can the authors prove that such obtained info can be used as GT. The last sentence of the first paragraph in Sec 4.1 is not clear enough for me.
7. In Fig, 4, I did not think the performance of the proposed model is better than the existing work, or in other words, we say close to GT. In the third scene, why did the proposed model predict one more trajectory?
8. The human study in Tab. 6 cannot convince me. I highly suggest the authors can provide explicit user study with clear numerical results instead of giving similar numbers with similar positive and negative ranges. Besides, please state the difference between the ones w/ and w/o mark. Furthermore, the use study without comparing to the existing works does not make sense to me.

---

> ### Author Response · Authors · 2025-07-16
> **author response**
>
> **Difference between the two tasks.** Thank you for asking for clarification of the two proposed tasks. The main difference lies in the distinct language instructions, i.e., the reasoning efforts required to get the correct answer. For the original task, e.g., Vanilla Hand Prediction Task (VHP), the prompts are simple and direct, specifying the action and object to execute. However, in the Reasoning-based Hand Prediction Task (RBHP), the labels of action and object are no longer at hand, where the intentions are implicitly described by a Vision-language Model(VLM). In this case, the agents are required to reason with the cues within the video.
>
> **Comment about writing.** Thank you for pointing out our problem with writing. We will make the descriptions clearer and easier to understand in the final version. Please let us know if there are any specific writing issues that the reviewer would like us to address.
>
> **Using existing techniques.** We are extending prior work on VLM to a new task of low-level hand prediction tasks, which opens up more possibilities compared to prior works on video visual-language models that cannot produce low-level hand trajectories and hand prediction tasks that do not support language grounding and reasoning.
>
> **Future horizon.** Thank you for asking about the detailed future horizon. As mentioned in section 5.1, the model is set to predict the hand position in the next 1 second at FPS = 4.
>
> **Hand tokenization.** Thank you for asking about the hand tokenization. The hand token is treated as a special token during the process. More specifically, we extend one more entry to the original vocabulary. The resulting hand embedding is the sum of the token embedding and the positional embedding of the 2D hand coordinate.
>
> **Generating GT labels.** Thank you for raising the problem. We acknowledge that using existing models for pseudo-GT generation may introduce some inaccuracies. However, automatic extraction is the most practical approach for large-scale Internet video data, and this methodology is widely adopted in prior works[1-2]. While pseudo-GT labels have inherent limitations, they provide sufficient supervision for effective model training.
>
> **Performance of prediction.** Thank you for this important observation. We clarify that our model's objective is not to perfectly replicate ground truth trajectories, but rather to generate realistic and plausible trajectory predictions. The apparent difference you noted in the third scene stems from our use of temperature sampling during inference, which introduces stochasticity into the prediction process. This means our model can produce multiple valid trajectory hypotheses rather than a single deterministic output.
>
> **Human Evaluation Study.** We thank the reviewer for the constructive feedback on our human evaluation study. We provide the following clarifications:
>
> The † symbol denotes HandsOnVLM fine-tuned on additional reasoning tasks (as mentioned in Section 5.1), while HandsOnVLM refers to the base model trained only on VHP datasets. We also would like to note that the human evaluation is to demonstrate the improved instruction-following ability on the reasoning-heavy prompts given by humans. We have shown that the fine-tuned model can generate trajectories with higher fidelity and match the language better. Regarding comparisons with existing work, conducting fair comparisons is challenging because they either do not predict language-grounded hand trajectory[1-2] or do not predict low-level actions for visual question-answering[3]. We are dedicated to including additional human evaluations in the revised version.
>
> [1] Bao, Wentao, et al. "Uncertainty-aware state space transformer for egocentric 3d hand trajectory forecasting." Proceedings of the IEEE/CVF International Conference on Computer Vision (2023): 13702-13711.
>
> [2] Liu, Shaowei, et al. "Joint Hand Motion and Interaction Hotspots Prediction from Egocentric Videos." Proceedings of the IEEE/CVF Conference on Computer Vision and Pattern Recognition (2022).
>
> [3] Huang, De-An, et al. "LITA: Language Instructed Temporal-Localization Assistant." arXiv preprint arXiv:2403.19046 (2024).

---

### Review · Reviewer_pB3k · 2025-06-30

**Summary Of Contributions:**

### The paper presents HandsOnVLM, a unified vision-and-language model that forecasts hand trajectories from short video clips and natural-language instructions.
---
For example, given the prompt “open a can,” a human’s steps might be:

-- Grasp the can with the left hand

-- Pick up the can opener with the right hand

-- Position the opener’s blade on the lid

-- Rotate the can while holding the opener steady

---

HandsOnVLM learns to predict exactly those motions in order, alongside any explanatory text.

## Visual Tokenization

A visual backbone processes T successive frames into T×M patch embeddings.

---

We have two types of tokens to consider
- Fast tokens – the per-frame average of M embeddings (one “fast” vector per frame) [For clarification It will be a single avergaed value over a frame's M token]
- Slow tokens – M spatial tokens obtained by uniformly sampling s frames and applying 2D average-pooling (SxS) over their M×M grids

---

## Vision-to-Language Fusion

Both fast & slow tokens are projected into the LLM’s embedding space and fused with the text query (“open a can”).
The combined embeddings serve as the context for autoregressive decoding.

## Autoregressive Decoding + CVAE Trajectory Head

During generation the model alternates between predicting words and a special "HAND" token.

Whenever "HAND" is emitted, its last-layer hidden state is fed into a lightweight CVAE decoder, which outputs the next 2D hand coordinate (e.g. the pixel location where the opener contacts the lid).
That coordinate is then re-embedded into the "HAND" slot so that all future predictions remain conditioned on the full motion history, preserving temporal and spatial consistency.

## Evaluation Benchmarks

Paper defines two new metric for the task.

---

Vanilla Hand Prediction (VHP): explicit “how-to” commands (e.g. “open a can”) → trajectory forecasting

Reasoning-based Hand Prediction (RBHP): implicit, commonsense queries (e.g. “How would I reach to turn on the light switch?”) → plausible hand motions

---


Paper uses existing methods and models to approach this unique problem statement.

**Audience:**

Yes

**Broader Impact Concerns:**

Not applicable.

**Claims And Evidence:**

Yes

**Requested Changes:**

- Need additional ablation study to justiy novel contribution (slow + fast token). If it's well establish method, please share the resource as i am not aware.

**Strengths And Weaknesses:**

## Strengths
- The paper is clearly and cleanly written, with intuitive figures and a well-organized appendix.
- It shows that integrating multiple off-the-shelf components yields strong end-to-end hand-trajectory forecasting on both explicit and reasoning tasks

## Weakness
- Incremental novelty, all the components are off-the-shelf.
- Niche task. The paper and methodology of approach seems very limited and ineffective to similar poblems.
- Missing ablation study/motivation for fast tokens. (Maybe its not needed?)

---

> ### Author Response · Authors · 2025-07-16
> **author response**
>
> **Incremental novelty.** We thank the reviewer for this feedback. While we acknowledge the reliance on off-the-shelf components, we argue that the value lies in the whole model and prediction task itself. Rather than reinventing established methods, we leverage proven components and focus our contribution on the novel proposed prediction tasks and the whole model as a system.
>
> **Niche task.** We thank the reviewer for this feedback. We respectfully disagree with the assessment that this is a niche task. Hand trajectory prediction has numerous applications across robotics, augmented reality, and human-computer interaction domains[1-5]. The methodology's scope extends well beyond the specific scenarios demonstrated, with clear potential for transfer to manipulation planning, AR interaction design[4], and assistive robotics[5]. In this paper, we extend hand prediction into the reasoning-required tasks, which is more natural and practical for humans to interact with. This task also has the potential to improve the important low-level reasoning ability of the visual-language models.
>
> **Slow-fast tokens.** Thank you for asking for the details of slow-fast tokens. Many previous works have suggested that the slow-fast tokens are beneficial for video understanding. In work[6], the authors have demonstrated the contribution of slow-fast tokens on both the video action classification and detection tasks. Recently, work[7] deployed a similar design into the video question-answering problems and achieved the state-of-the-art performance on video understanding and answering tasks.  We are doing this additional experiment now as suggested, and will add the results in the revised version.
>
> [1] Shikhar Bahl, Russell Mendonca, Lili Chen, Unnat Jain, and Deepak Pathak. "Affordances from human videos as a versatile representation for robotics." Proceedings of the IEEE/CVF Conference on Computer Vision and Pattern Recognition (2023): 13778-13790.
>
> [2] Anthony Brohan, Noah Brown, Justice Carbajal, Yevgen Chebotar, Xi Chen, Krzysztof Choromanski, Tianli Ding, Danny Driess, Avinava Dubey, Chelsea Finn, et al. "RT-2: Vision-language-action models transfer web knowledge to robotic control." arXiv preprint arXiv:2307.15818 (2023).
>
> [3] Homanga Bharadhwaj, Debidatta Dwibedi, Abhinav Gupta, Shubham Tulsiani, Carl Doersch, Ted Xiao, Dhruv Shah, Fei Xia, Dorsa Sadigh, and Sean Kirmani. "Gen2act: Human video generation in novel scenarios enables generalizable robot manipulation." arXiv preprint arXiv:2409.16283 (2024).
>
> [4] B. Schwald, B. Seibert, and T. Weller. "A hand-interaction model for augmented reality enhanced human-robot collaboration." CIRP Annals - Manufacturing Technology 73, no. 1 (2024): 5-8.
>
> [5] Yujiao Cheng, Liting Sun, Changliu Liu, and Masayoshi Tomizuka. "Towards efficient human-robot collaboration with robust plan recognition and trajectory prediction." IEEE Robotics and Automation Letters 5, no. 2 (2020): 2602-2609.
>
> [6] Christoph Feichtenhofer, Haoqi Fan, Jitendra Malik, and Kaiming He. SlowFast Networks for Video Recognition. In Proceedings of the IEEE/CVF International Conference on Computer Vision (ICCV), pages 6202-6211, 2019.
>
> [7] Huang, De-An, et al. "LITA: Language Instructed Temporal-Localization Assistant." arXiv preprint arXiv:2403.19046 (2024).

---

### Review · Reviewer_tiy7 · 2025-07-02

**Summary Of Contributions:**

This paper proposes HandsOnVLM, a video-based VLM for generating textual responses and predicting future hand motion from ego-centric video contexts given language prompts. The model casts hand trajectory prediction as auto-regressive next token prediction conditioned on fused video and language tokens. To consider different forms of language conditions, the traditional hand prediction tasks are extended to two new tasks: Vanilla Hand Prediction (VHP) and Reasoning-based Hand Prediction (RBHP). Experiments show that HandsOnVLM outperforms baselines on both VHP and RBHP tasks, demonstrating strong generalization and reasoning capabilities.

**Audience:**

Yes

**Claims And Evidence:**

Yes

**Requested Changes:**

- Add the mentioned open-source resources.

- Revise the above writing issues.

- Results for fine-tuning also the visual backbone.

**Strengths And Weaknesses:**

Strengths

++ The new extended tasks VHP and RBHP enable hand trajectory prediction from ego-centric videos based on different forms of language queries.

++ The method demonstrates strong generalization and reasoning abilities on diverse real-world datasets.


Weaknesses:

-- "... we open-source to the community ...": Where can we find the mentioned open-source resources?

-- Section 3.1: The explanation regarding the extraction of slow tokens is inaccurate.

-- The visual backbone is frozen for this task. How would the model perform if the visual backbone is also fine-tuned?

-- How are the hand trajectories tokenized? Are they tokenized using the employed LLM's native tokenizer?

-- "plauisble" in Figure 1 is a typo.

-- page 2: "and and hand"

-- The notation of lamba_{txt} in page 4 is inconsistent with that in Eq. (1).

---

> ### Author Response · Authors · 2025-07-16
> **author response**
>
> **Open-source resources.** Thank you for asking about the open-source resources. Our code is open-sourced but since TMLR requires preserving anonymity, we did not link it in the paper. We will be happy to provide the reference to it if we are allowed to link it. For now to preserve anonymity, we have also uploaded the code to an open-source bucket https://github.com/HandsOnVLM-release/HandsOnVLM .
>
> **Explanation of the slow token.** Thank you for requesting a clearer explanation of slow tokens. Building on prior research showing that video frames often contain redundant information, we implement fast-slow token compression to enhance model efficiency. Slow tokens are specifically designed to capture spatial information, while fast tokens focus on extracting temporal dynamics.
> Given an original video with $T$ \times $M$ total tokens, we uniformly sample $s^2$ frames and apply spatial pooling at ratio $s$. This process yields $M$ spatial tokens that encode the essential spatial characteristics of the video.
>
> **Frozen backbone.** Thank you for suggesting adding the unfrozen experiments. We follow the standard practice of training the Vision-language Model, freezing the visual backbone during training. Many previous works[1, 2, 3] have demonstrated the effectiveness of this method. Previous works also found that fine-tuning will greatly harm the reasoning ability. For example, Prismatic VLMs[1] found that fine-tuning the visual backbone significantly degrades performance, especially on tasks requiring fine-grained spatial reasoning. We are doing this additional experiment now as suggested, and will add the results in the revised version.
>
> **Hand tokenization.** Thank you for asking about the hand tokenization. The hand token is treated as a special token during the process. More specifically, we extend one more entry to the original vocabulary. The resulting hand embedding is the sum of the token embedding and the positional embedding of the 2D hand coordinate.
>
> **Miscs.** We thank the reviewer for carefully reading our manuscript and identifying these errors. We will correct all of them in the revised version.
>
> [1] Liu, Haotian, et al. "Visual instruction tuning." Advances in Neural Information Processing Systems 36 (2023): 34892-34916.
>
> [2] Feng, Yao, et al. "Chatpose: Chatting about 3d human pose." Proceedings of the IEEE/CVF Conference on Computer Vision and Pattern Recognition (2024): 2093-2103.
>
> [3] Huang, De-An, et al. "Lita: Language instructed temporal-localization assistant." European Conference on Computer Vision. Springer, 2024. 202-218.
>
> [4] Karamcheti, Siddharth, et al. "Prismatic vlms: Investigating the design space of visually-conditioned language models." Forty-first International Conference on Machine Learning. 2024.

---

### Decision · Action_Editor_Yqp2 · 2025-08-01

**Recommendation:** Accept with minor revision

**Additional Comments:**

A revision has not yet been submitted.  The revision will need to provide the promised updates, fixes, and clarifications.

**Audience:**

Yes

**Audience Explanation:**

Forecasting in general is an interesting and open problem, and hand trajectory forecasting could be useful for a variety of robotics and egocentric understanding applications.

**Claims And Evidence:**

Yes

**Claims Explanation:**

All reviewers support accepting the paper to TMLR.  Generally, the reviews indicate that the contributions are well supported by evidence.  Most requests/comments are clarifications, and requests for additional ablations/experiments are addressed in the rebuttal.